# Hepatitis C Virus Enhances the Invasiveness of Hepatocellular Carcinoma via EGFR-Mediated Invadopodia Formation and Activation

**DOI:** 10.3390/cells8111395

**Published:** 2019-11-05

**Authors:** Liat Ninio, Abraham Nissani, Tomer Meirson, Tom Domovitz, Alessandro Genna, Shams Twafra, Kolluru D. Srikanth, Roba Dabour, Erez Avraham, Ateret Davidovich, Hava Gil-Henn, Meital Gal-Tanamy

**Affiliations:** 1Molecular Virology Laboratory, Azrieli Faculty of Medicine, Bar-Ilan University, Safed 1311502, Israel; liatus@gmail.com (L.N.); abraham.nissani@gmail.com (A.N.); tomdo17@gmail.com (T.D.); roba.da1383@gmail.com (R.D.); erezbya@walla.co.il (E.A.); Ateret.Davidovich@biu.ac.il (A.D.); 2Cell Migration and Invasion Laboratory, Azrieli Faculty of Medicine, Bar-Ilan University, Safed 1311502, Israel; tomermrsn@gmail.com (T.M.); genna.ale@gmail.com (A.G.); shams_a_s@windowslive.com (S.T.); dutt415@gmail.com (K.D.S.); 3Drug Discovery Laboratory, Azrieli Faculty of Medicine, Bar-Ilan University, Safed 1311502, Israel

**Keywords:** hepatitis C virus, hepatocellular carcinoma (HCC), invasion, metastasis

## Abstract

Hepatocellular carcinoma (HCC) represents the fifth most common cancer worldwide and the third cause of cancer-related mortality. Hepatitis C virus (HCV) is the leading cause of chronic hepatitis, which often results in liver fibrosis, cirrhosis, and eventually HCC. HCV is the most common risk factor for HCC in western countries and leads to a more aggressive and invasive disease with poorer patient survival rates. However, the mechanism by which the virus induces the metastatic spread of HCC tumor cells through the regulation of invadopodia, the key features of invasive cancer, is still unknown. Here, the integration of transcriptome with functional kinome screen revealed that HCV infection induced invasion and invadopodia-related gene expression combined with activation of host cell tyrosine kinases, leading to invadopodia formation and maturation and consequent cell invasiveness in vitro and in vivo. The promotion of invadopodia following HCV infection was mediated by the sustained stimulation of epidermal growth factor receptor (EGFR) via the viral NS3/4A protease that inactivates the T-cell protein tyrosine phosphatase (TC-PTP), which inhibits EGFR signaling. Characterization of an invadopodia-associated gene signature in HCV-mediated HCC tumors correlated with the invasiveness of HCC and poor patient prognosis. These findings might lead to new prognostic and therapeutic strategies for virus-mediated invasive cancer.

## 1. Introduction

Hepatocellular carcinoma (HCC) is the fifth most common cancer worldwide. Advanced liver fibrosis is the most prominent feature associated with higher HCC risk. HCC is the third cause of cancer-related mortality, with a five-year survival rate of less than 10%. The major reasons for high mortality rates are disease aggressiveness and lack of effective treatment options [1]. The extremely poor prognosis of HCC is a result of a high recurrence rate of intrahepatic metastasis that is developed by the invasion of the cells from the primary tumor into other parts of the liver or by their dissemination via the portal vein. Metastasis occurs in more than 50% of patients after surgery, which remains the most effective treatment for early localized tumors [2]. Mono-therapy with sorafenib, a multikinase inhibitor, prolongs overall survival for several months and delays the tumor progression in patients with advanced HCC [3]. Studies suggest that aggressive treatment may result in earlier extrahepatic metastasis, despite a less advanced intrahepatic tumor stage [4]. Thus, as the malignant potential of intrahepatic tumors is associated with extrahepatic spread and survival, a combination of intrahepatic local treatment and metastasis prevention may be useful. However, such effective and well-tolerated treatments for advanced HCC are absent, highlighting the need for new therapeutic approaches.

Metastatic cancer cells must penetrate through several barriers in order to escape the primary tumor and gain entry into the bloodstream for disseminating into other tissues. Invasive cancer cells penetrate these barriers by forming invadopodia—F-actin rich protrusions that localize matrix-degrading activity to cell-substrate contact points and represent sites in which cell signaling, proteolytic, adhesive, cytoskeletal, and membrane-trafficking pathways physically converge [5]. Invadopodia were identified in several invasive cancer cell lines, such as breast, head and neck, prostate, fibrosarcoma, and melanoma [6], as well as in primary tumor cells [7]. Recent evidence also demonstrates direct molecular links between invadopodia assembly and function and metastasis in mice models [8] and human patients [9].

Hepatitis C virus (HCV) is the most common risk factor for HCC in western countries [10] and a major public health problem with over 71 million infected people worldwide who are at risk of developing a life-threatening liver disease [11]. Despite the recent approval of direct-acting antiviral drugs (DAAs), many challenges, such as limited implementation, efficacy, and protection from re-infection, are expected to delay the decrease in HCC incidence [12]. Moreover, multiple studies have reported that sustained virological responses (SVR) do not eliminate cancer development [13], and increased recurrence of cancers following HCV treatment has also been reported [14]. Several epidemiological and clinical studies suggest that HCV-related HCC shows more aggressive tumor features, higher tumor multifocality and vascular invasion, and increased recurrence rate following curative resection [15,16,17]. Although the recent report has demonstrated that HCV replicates at lower levels in HCC cells, still, replication in these cells was documented [18], pointing for possible direct viral-mediated mechanisms of enhancement of invasion in HCC cells. Indeed, recent evidence suggests that HCV induces activation of migration and invasion pathways [19,20,21,22,23,24,25,26,27,28,29,30,31,32,33]. These reports demonstrate that viral proteins (core, envelope, NS3, NS5A, and NS4B) affect the VEGF, TGF-β, and Wnt/β-catenin signaling cascades to activate the epithelial-to-mesenchymal transition (EMT) process that is known as an early event occurring during the processes of cancer metastasis. However, the molecular and cellular mechanisms by which HCV mediates intra and extrahepatic dissemination of HCC cells through regulation of invadopodia, the key features of invasive cancer, and its effect on cell motility and metastasis in vivo, have not been demonstrated before. Strikingly, we have recently reported that while HCV-induced genomic instability events following cure of HCV with DAAs are reversed [34], epigenetic, and the consequent gene expression, alterations induced by HCV remain as an epigenetic signature following the cure. Importantly, the most significant pathways that remain persistently altered are cytoskeleton remodeling pathways that are known to play a part in invadopodia formation and invasion [35].

Here, we used both functional and genomic screens to systematically elucidate the molecular mechanisms that contribute to HCV-induced invasiveness of HCC. We demonstrated that HCV infection activated the epidermal growth factor receptor (EGFR) signaling pathway, thus promoting invadopodia formation and function. Consequently, the invasiveness of infected liver cancer cells was enhanced, leading to significantly higher intra and extrahepatic metastatic dissemination, compared to non-viral HCC. We presented a novel mechanism by which HCV infection led to higher invasiveness and aggressiveness of HCC tumors and, consequently, to poorer patient outcomes.

## 2. Materials and Methods

### 2.1. Constructs and Cell Lines

The Huh7.5 cell line (a generous gift from Dr. Charles M. Rice, The Rockefeller University, New York, NY, USA) and the Huh7/FT3-7 cell line (a generous gift from Dr. Stanley M. Lemon, UNC-Chapel Hill, Chapel Hill, NC, USA) are human hepatocellular carcinoma cell lines that are highly permissive for replication of sub-genomic and full-length HCV RNA [36].

pCDNA3.1-Hygro-NS3/4A and pCDNA3.1-Hygro-Core for the expression of NS3/4A or Core, respectively, were generous gifts from Prof. Itai Benhar, Tel-Aviv University, Tel-Aviv, Israel. p3X-Flag-CMV-7.1 (Sigma) and the cDNA for human T-cell protein tyrosine phosphatase (TC-PTP) were kindly provided by Dr. Georg Bode, Department of Gastroenterology, Hepatology, and Infectiology, Heinrich-Heine-University, Düsseldorf, Germany. The cDNA of human TC-PTP was amplified and subcloned into the p3X-Flag-CMV-7.1 expression vector.

### 2.2. Virus Infection

Cell culture infectious HCV HJ3-5 chimeric virus [37] stocks were produced in Huh7/FT3-7 cells followed by viral titer determination using the focus forming units (FFU) assay in Huh7.5 cells, as previously described [37]. For all experiments, Huh7.5 cells were infected with the virus at a multiplicity of infection (MOI) of 0.1 or 0.5 and passaged for two weeks until approximately 100% of the cells were HCV positive, as we previously described [38].

### 2.3. Transfections

Plasmids (pCDNA3.1-Hygro-NS3/4A/pCDNA3.1-Hygro-Core/p3X-Flag-CMV-7.1/pCMV control) were transfected into Huh7.5 cells or HCV-infected Huh7.5 cells using Lipofectamine 2000 (Invitrogen, Carlsbad, CA, USA) according to manufacturer’s instruction. Transfection efficiency was determined by immunostaining with HCV-positive serum of infected subject, anti-TC-PTP (R&D Systems, Minnesota, MN, USA) antibody, anti-core (Thermo Fisher Scientific, Waltham, MA, USA) antibody, and anti-human 488 Alexa fluor (Jackson, West Grove, PA, USA) as the secondary antibody.

Huh7.5 cells (2 × 10^6^) transfected with NS3/4A or pCMV control vector were treated for 24 h with 1 μM erlotinib (LC labs, Woburn, MA, USA) before measuring invadopodium formation. Cell viability following erlotinib treatment was measured using the XTT kit II (Roche Life Science, Penzberg, Germany) according to the manufacturer’s instructions.

MT1-MMP (MMP14) siRNA (ON-TARGETplus human mmp-14 siRNA nm_004995) and negative control siRNA (ON-TARGETplus non-targeting pool) were purchased from Dharmacon (Lafayette, Colorado, United States). HCV-infected and non-infected cells were transfected with 6 μg of siRNA using Oligofectamine Reagent (Invitrogen), according to the manufacturer’s instructions, and used for matrix degradation assay, as described below.

### 2.4. Scratch Wound Assay

HCV-infected and non-infected Huh7.5 cells were plated in 6 wells plates at 95–100% confluence. At 24 h following plating, cells were washed with PBS, and mitomycin was added to the monolayer for 2 h to inhibit cell proliferation. Then, the monolayer was wounded using a sterile pipette tip. The plate was placed in a 37 °C heated chamber, and images were collected using the LSM780 inverted microscope (10×, NA 0.25, air objective). Phase images were acquired every hour for 53 h. Wound closure was analyzed using ImageJ software (version 1.51n, NIH, Bathesda, MA, USA).

### 2.5. Invadopodium Precursor Formation Assay

Non-infected, HCV-infected, or Huh7.5-transfected cells were stained for invadopodium precursors as punctate structures in which F-actin and cortactin co-localize, as previously described [39]. In brief, cells were plated on MatTek glass-bottom dishes coated with AlexaFluor 405/488–conjugated gelatin and incubated at 37 °C for 24 h. Cells were then fixed with 3.7% formaldehyde/PBS for 20 min and then permeabilized with 0.1% Triton X-100 in PBS for 5 min at room temperature. After three washes, cells were incubated in blocking solution 1%BSA/1%FBS/PBS, labeled with primary antibodies cortactin (Abcam, Cambridge, UK) for 2 h, and followed by labeling with secondary fluorochrome-conjugated antibodies for 1 h. Actin filaments were visualized with Alexa-phalloidin (Invitrogen). Cells were imaged with a Life-imaging microscope (Zeiss). Invadopodia were detected by the co-localization of cortactin and actin. The number of invadopodia per cell was counted using ImageJ software.

### 2.6. Transwell Migration and Invasion Assay

Chemotactic migration and invasion were measured using the Transwell assay, as previously described [40]. In brief, for migration assay, HCV-infected and non-infected Huh7.5 cells were seeded in transwell inserts (BD, San Jose, CA, USA) at 80–90% confluence and allowed to invade for 24 h towards the medium. At the end of the incubation period, cells that were invaded through the membrane of the insert were stained with crystal violet and counted. Invasion assay was performed as above, but the transwell inserts were coated with Matrigel (BD) before the cells were seeded.

### 2.7. Matrix Degradation Assay

Matrix degradation assays were performed using Huh7.5 cells cultured on gelatin matrix, which was prepared as described previously [41]. In brief, Alexa Fluor 405/488–conjugated gelatin was prepared by labeling porcine gelatin (Sigma) with Alexa Fluor 405/488 (Invitrogen) according to the manufacturer’s instructions. MatTek glass-bottom dish was pre-cleaned with 1N chloride acid for 10 min, followed by extensive washing with PBS. The MatTek glass-bottom dishes were coated with 50 mg/mL poly-l-Lysine (Sigma) for 20 min at room temperature, washed with PBS, and fixed with 0.5% glutaraldehyde (Sigma) for 15 min followed by extensive washing. The glass-bottom dishes were then covered with a drop of gelatin matrix (0.2% gelatin and AlexaFluor 405/488–labeled gelatin at 1:20 ratio) and incubated for 10 min at room temperature. After washing with PBS, dishes were incubated with 5 mg/mL sodium borohydride for 15 min, sterilized with ethanol for 15 min and washed three times in PBS, and finally incubated in 2 mL of DMEM for 30 min before adding the cells. To examine the ability of cells to degrade matrix, 10^4^ cells were plated on MatTek glass-bottom dishes coated with AlexaFluor 405/488 and incubated at 37 °C for 72 h. Cells were then fixed with 3.7% formaldehyde/PBS for 20 min, washed with PBS, and labeled with Hoechst (Sigma). Cells were imaged by a Life-imaging microscope (Zeiss). Foci of the degraded matrix were visible as dark areas of 0.2 to 1.2 Am in diameter, lacking fluorescence and appearing as “holes” in the bright fluorescent gelatin matrix. A cell degrading at least one hole under the cell or near the cell edge was counted as a cell able to degrade the matrix. Degradation was analyzed by quantifying the average degraded area pixels per field using ImageJ software and normalized to the number of cells in each field.

### 2.8. RNA-Seq

Total RNA from HCV-infected and non-infected Huh7.5 cells was purified using the RNeasy Mini Kit (Qiagen, Hilden, Germany). One microgram of total RNA was treated with the NEB Next poly (A) mRNA Magnetic Isolation Module (NEB, Ipswich, MA, USA). RNA-seq libraries were produced by using the NEB Next Ultra RNA Library Prep Kit for Illumina (NEB) and sequenced by Illumina HiSeq 2500 platform with single-end reads of 50 bp according to the manufacturer’s instructions.

### 2.9. RNA-Seq Bioinformatical Analysis

Short sequence tags were quality trimmed, discarding sequences shorter than 35 nt. The sequences were aligned to the GRCh37 (hg19) reference genome using STAR [42]. Aligned reads were counted using FeatureCounts [43], by means of the Ensemble Version 75 transcript database. Differential analysis was performed using edgeR [44].

### 2.10. RNA Extraction and Quantitative Real-Time PCR (qRT-PCR)

Total RNA was extracted from HCV-infected and non-infected Huh7.5 cells using the RNeasy kit (Qiagen). Equal amounts of total RNA were reverse transcribed using the high capacity cDNA reverse transcription kit (Applied Biosystems, Foster City, CA, USA). Oligonucleotides were designed using the Primer3 PCR Primer Design Tool. The qRT-PCR analysis was performed using the Power SYBR Green Master Mix (Life Technologies, Carlsbad, CA, USA) and gene sequence-specific primers (Appendix A).

Alternatively, genes were amplified using the gene-specific PrimeTime^®^ Std qPCR Assay (Integrated DNA Technologies, Coralville, IA, USA). The thermal program included 10 min incubation at 95 °C followed by 40 cycles of 95 °C for 10 s, 58 °C for 10 s, and 72 °C for 20 s. The fluorescence readings were recorded after each 72 °C step. Dissociation curves were performed after each PCR run to ensure that a single PCR product had been amplified. Differential expression was calculated using the equation: 2^(−ΔΔCt)^, with *GAPDH* as a housekeeping gene control.

### 2.11. Bioinformatics Analysis

Comparative literature mining was performed using two different automated literature mining tools, Gene List Automatically Derived for You (GLAD4U) (http://bioinfo/vanderbuilt.edu/glad4u/) and Agilent Literature Search (ALS) (http://apps.cytoscape.org/apps/agilentliteraturesearch). GLAD4U search was performed using the query “invadopodia” or “invasion” and limited to the human context with a threshold of 0.01. ALS search was performed through Cytoscape using the query “invadopodia” or “invasion” and limited to *Homo sapiens* with limited interaction lexicon.

The combined lists of proteins identified by GLAD4U and ALS (a total of 425 or 1066 non-redundant proteins for invadopodia or invasion, respectively) and the list of proteins identified in mRNA-seq screen with positive fold change and *p*-value <0.05 (a total of 1753 non-redundant proteins) or the list of proteins obtained in upstream kinase analysis was used to build networks of physical and functional associations in STRING (http://www.string-db.org). Networks were overlayed in Cytoscape (http://www.cytoscape.org). GeneCards (http://www.genecards.org) and UniProt (http://www.uniprot.org) databases were used to classify overlapped proteins and ligands with their cognate receptors by manual curation.

### 2.12. Western Blot

At the end of experimental treatment, cells were washed with PBS and solubilized in RIPA lysis buffer, with freshly added 1% protease inhibitor cocktail and 1 mM phenylmethylsulfonyl fluoride (PMSF). Protein concentration was estimated using the BioRad protein assay. Samples were analyzed by western blot. Samples were separated by SDS-PAGE and transferred onto nitrocellulose membranes. After blocking in 5% skim milk, the blots were incubated with primary antibodies: cortactin (Abcam), TC-PTP (R&D Systems), core (Thermo Fisher Scientific) or positive serum for HCV-infected patient, and β actin (Sigma) overnight at 4 °C and then with HRP-conjugated secondary antibody for 1 h. Reactive bands were visualized with ECL reagent (Pierce, Rockford, IL, USA) using the LAS documentation system.

### 2.13. Upstream Kinase Analysis

HCV-infected and non-infected Huh7.5 cells were washed with PBS and solubilized in M-PER mammalian protein extraction reagent (Thermo Fisher Scientific). Protein concentrations were estimated by using the BioRad protein assay. Ten micromolar of protein lysates were subjected to a functional kinome screen by using Tyrosine Kinase PamChip Array for PamStation12 that consists of 144 peptides with known phosphorylation sites. Each peptide represents a 13- to 15-amino-acid sequence corresponding to a phosphorylation site, which serves as a kinase substrate. Together, these peptides are predicted to cover the activity of around 65% of the human kinome. The assay was performed according to the manufacturer’s protocol and as previously described [18]. Upstream kinase data analysis was performed using BioNavigator software (PamGene, Hertogenbosch, The Netherlands).

### 2.14. Mouse Xenograft Model

All experimental procedures were conducted in accordance with the Federation of Laboratory Animal Science Associations (FELASA) and were approved by the Bar-Ilan University animal care and use committee in Israel #27-04-2015. A total of 1 × 10^6^ Huh7.5 cells infected with HCV HJ3-5 virus or non-infected control cells were administered into 6-week-old NOD-SCID-gamma (NSG) male mice by intrahepatic injection into the left lobes of the liver. At four weeks following tumor cell injection, an IRDye 800CW 2-DG optical probe (LI-COR Biosciences, Lincoln, NE, USA) was injected into the tail vein of tumor-bearing mice. Mice were imaged using the Pearl Trilogy small animal imaging system (LI-COR Biosciences). Fluorescent signals from acquired images were analyzed using the Image Studio Lite software (LI-COR Biosciences). At 2 and 4 weeks following injection, the left lobe and right lobes of the liver as well as lungs from each mouse were harvested, and DNA or RNA was extracted for further analyses.

### 2.15. Quantification of Human DNA or RNA in Mouse Tissues

Quantification of human tumor cell DNA in mouse liver and lungs was performed by quantitative real-time PCR, as previously described [45]. Tissues were homogenized in urea lysis buffer (2% (*w*/*v*)) SDS, 10 mM EDTA, 0.35 M NaCl, 0.1 M Tris–HCl, pH 8.0, 7 M urea). Genomic DNA was extracted with the standard phenol–chloroform method and diluted in ultra-pure water. Real-time PCR primers were designed by using the Universal ProbeLibrary Assay Design Center tool (Roche Applied Science, Penzberg, Germany) and purchased from Sigma-Aldrich Chemie GmbH (Buchs, Switzerland). Human (Hu)-specific primers (forward: 5′-ctgttttgtggcttgttcag-3′, reverse: 5′-aggaaaccttccctcctcta-3′) amplified a 122-bp fragment of a region located in 7p15–p12. Mouse (Mo)-specific primers (forward: 5′-ttggttgagaagcagaaaca-3′, reverse: 5′-cacacagtcaagttcccaaa-3′) amplified a 181-bp fragment of a region located in 2F1–F3. Target sequences in the human and mouse genomes represent species-specific, unique, and conserved regions located in the non-transcribed regions of the human β-actin and mouse β-2-microglobulin genes, respectively. Real-time PCR was carried out in a StepOnePlus Real-Time PCR (Applied Biosystems) using the SYBR-Green PCR Master Mix (Applied Biosystems) according to the manufacturer’s protocol with 25 ng of total genomic DNA. The percent of human cells in the mice tissues was calculated.

To quantify the expression of specific genes in human cells that invaded into the mouse liver, total RNA was extracted from the left and right lobes of mouse livers using the RNeasy Mini kit and RNase-Free DNase Set. Human-specific primers were designed using the Primer3 PCR Primer Design Tool. The qRT-PCR analysis was performed as described above.

### 2.16. Immunohistochemistry

To detect human hepatocellular carcinoma in mouse tissues, liver and lung tissues were harvested and embedded in paraffin. Samples were sectioned at 5 µm and stained with hematoxylin/eosin (H&E) and with anti-human HLA (Santa Cruz, Dallas, TX, USA). Slides were observed and imaged using the Axio scan Z1 slide scanner (Zeiss).

### 2.17. Patient Survival Analysis

RNA sequencing data for 377 hepatocellular carcinoma samples were obtained from The Cancer Genome Atlas (TCGA, https://cancergenome.nih.gov/) [46] and analyzed with cBioPortal tools (http://www.cbioportal.org) [47] and MATLAB (The MathWorks, Inc., Natick, MA, USA). The analyzed dataset contained mRNA expression Z-scores (RNA-Seq V2 RSEM) computed as the relative expression of an individual gene and tumor to the expression distribution of all samples that are diploid for the gene. The tumor profiles represented HCC sampled at the time of surgical resection, annotated with disease-free survival (DFS) and overall survival (OS) time and censorship status. To visualize the hazard ratio (HR) distance correlations, we employed the cluster gram function in MATLAB using ward linkage for the hierarchical clustering. All statistical tests were two-sided.

### 2.18. Statistical Analysis

Statistical significance was calculated using unpaired, two-tailed Student’s *t*-test. Values were considered statistically significant if the *p*-value was ≤0.05. Error bars represent SEM.

## 3. Results

### 3.1. HCC Cell Invasion, but not Migration, is Enhanced by HCV Infection

In order to initiate the metastatic process, cancer cells must acquire the ability to migrate and invade. Indeed, it was previously demonstrated that HCV infection in cultured cells induces EMT and, consequently, enhancement of cell motility and invasion [19,20,21,22,23,24,25,26,27,28,29,30,31,32,33]. To gain more insight into these processes, we first performed the scratch wound assay with HCV-infected and non-infected Huh7.5 HCC cells, which evaluates two-dimensional (2D) cell motility. The Huh7.5 cells were infected with chimeric 1a/2a virus HJ3-5, which replicates and produces infectious viruses. Two weeks following virus infection, approximately 100% of the cells were positive for HCV (Appendix A) and then used for the subsequent experiments. To control for differences in growth rate between non-infected and HCV-infected cells during the time of the experiment, we performed the assay in the presence of mitomycin C that prevents cell proliferation. As demonstrated in Figure 1A,B, no differences in 2D motility of non-infected versus HCV-infected cells were observed. This observation is in contrast to previous studies, demonstrating that expression of HCV proteins or HCV infection increases cell migration [24,26,27,28,29,31,32,33,48], and may be related to the use of different liver cell lines or the lack of using mitomycin C for eliminating the possibility of differences in cell proliferation. To validate these results, we also performed a Transwell migration assay. We plated serum-starved HCV-infected or non-infected Huh7.5 HCC cells on Transwell membranes and measured their ability to migrate towards the complete medium. We observed a higher number of migrated cells in HCV-infected compared to non-infected cells; however, the difference was not statistically significant (Appendix A).

To test the possible effect of HCV on the chemotactic invasion of infected cells, we plated serum-starved HCV-infected or non-infected Huh7.5 HCC cells on Matrigel-coated Transwell membranes and measured their ability to invade towards the complete medium. As demonstrated in Figure 1C,D, HCV-infected cells showed a significant increase of approximately 6 fold in chemotactic invasion compared with non-infected cells. These data suggested that HCV significantly enhanced tumor cell invasion, but not 2D migration, in infected cells (Appendix A). This result is in line with our recent publication, demonstrating that 2D migration and extracellular matrix-dependent invasion are two different processes that are controlled by different sub-cellular cytoskeletal structures and signaling mechanisms within cancer cells [49].

We have recently demonstrated that infection with HCV induces epigenetic alterations that reprogram the host gene expression pattern that is implicated in the development of HCC. Analysis of RNA sequencing (RNA-seq) and chromatin immunoprecipitation sequencing (ChIP-seq) data from HCV-infected versus non-infected cells revealed cytoskeletal remodeling and invasiveness as the most significantly enriched signaling pathways [35]. The RNA-seq analysis of differential peaks (HCV-infected compared to non-infected cells) identified 2672 differentially expressed genes (DEGs) (using a threshold of *p* < 0.05, Log2FC ≥ 1.5, and Log2FC ≤ –1.5). Of these, HCV infection-induced up-regulation of 1865 genes (Log2 Fold Change > 1.5) [35]. To investigate whether altered gene expression by HCV leads to the enhanced invasiveness of infected cells, we used literature mining tools to prepare an integrated gene list of invasion-related genes that were intersected with HCV-induced up-regulated genes that were identified by RNA-seq. Using this approach, we identified 1066 invasion-associated genes; of these, 115 genes were overlapping between the two groups (Figure 1E, top). To determine if the overlap among the up-regulated genes were significantly enriched compared to a random group of genes, we calculated the cumulative probability of the hypergeometric distribution. The *p*-values of the invasion groups were 6.3 × 10^−10^, indicating a significant enrichment of invasion-associated genes.

Manual classification of the overlapping genes identified biological processes and protein groups that were directly related to invasion, such as cytoskeletal and non-cytoskeletal signaling, cell adhesion, extracellular matrix proteins, matrix metalloproteinases (MMPs), and cytoskeletal scaffold proteins, as well as genes involved in regulation of gene transcription (Figure 1E, bottom and Appendix A). Together, these observations suggested that HCV infection led to cancer cell invasiveness by enhancing the expression of invasion-associated genes.

### 3.2. HCV Infection Induces the Expression of Invadopodia-Associated Genes

Because invadopodia are a key feature of invasive cancer cells and have been linked to invasiveness in both cultured cells and mice models, we hypothesized that HCV induces invasiveness by regulating the formation and/or activation of invadopodia in infected cells. To test this hypothesis, we first repeated our literature mining analysis using an integrated list of invadopodia-related genes that were intersected with the 1865 HCV-induced up-regulated genes identified by RNA-seq. Using this approach, we identified 425 invadopodia-related genes; among these, 56 overlapped between the two groups (Figure 2A, top). The cumulative probability of the hypergeometric distribution provided *p*-values of 1.6 × 10^−8^, indicating a significant enrichment of invadopodia-associated genes.

Manual classification of the overlapping genes identified biological processes and protein groups that were directly related to invadopodia, such as signal transduction and cytoskeletal signaling, actin remodeling, vesicular transport, growth factors and cytokines, matrix metalloproteinases, as well as genes involved in regulation of gene transcription (Figure 2A, bottom and Appendix A).

To validate the induced expression of invadopodia-associated genes by HCV, we performed qRT-PCR on mRNA samples isolated from HCV-infected and non-infected cells. Huh7.5 cells were infected with (approximately 100% infection, prepared as described in the methods), and the efficiency of infection was validated by measuring virus RNA levels in infected and non-infected cells (Figure 2B). Our qRT-PCR analysis revealed a significant enhancement of all tested invadopodia-related gene expression following HCV infection, except for gene *SH3PXD2A* (TKS5) (Figure 2C). Since cortactin is a marker for invadopodia formation as an essential scaffold protein of invadopodia (expressed by *CTTN* gene), we also validated the increase in cortactin protein following HCV infection (Figure 2D).

Overall, these data demonstrated that HCV infection up-regulated the expression of multiple invadopodia-associated genes and implied that this orchestrated change in gene expression led to increased cancer cell invasiveness.

### 3.3. Infection with HCV Enhances Invadopodium Precursor Formation and Activation in HCC Cells

The alteration of the gene expression pattern following HCV infection points to the misregulation of invadopodia formation and function. To validate the effect of HCV infection on the initial assembly of invadopodium precursors, HCV-infected (100% infected as detected by immunostaining for viral proteins) and control non-infected HCC cells were plated on gelatin matrix and labeled for the invadopodium precursor markers actin and cortactin (Figure 3A, left). These markers function as regulatory and structural components in the invadopodia assembly process. As actin-based structures, invadopodia contain a primarily branched F-actin core. Cortactin is an actin-binding protein and an essential scaffold protein of invadopodia. We quantified the co-localization of actin and cortactin that represent invadopodia in the cells, as was previously described [41].

As demonstrated in Figure 3A (right), HCV-infected cells showed a significant increase in punctuated actin- and cortactin-positive staining. By counting the overlapping staining pointing for invadopodium precursors, we observed a significant increase in their number in HCV-infected compared to non-infected cells.

At their final stage of maturation, invadopodia acquire the ability to focally degrade the extracellular matrix via recruitment and activation of MMPs [50]. To examine the effect of HCV infection on MMP-mediated extracellular matrix degradation by mature invadopodia, HCV-infected and non-infected cells were plated on a fluorescently labeled gelatin matrix (Figure 3B, left). Quantification of the degradation area revealed a significant increase in the ability of HCV-infected cells to degrade the matrix compared to non-infected cells (Figure 3B, right).

One of the HCV-induced genes that were up-regulated in our RNA-seq analysis and is directly related to invadopodia is *MMP14*, which encodes the transmembranal metalloproteinase MT1-MMP (Figure 2C). Evaluation of the change in *MMP14* expression compared to other invadopodia-activated MPPs, *MMP2* and *MMP9*, showed a significant increase in expression of all three MMPs; however, *MMP14* showed the highest increase in expression following infection (Appendix A). To investigate whether the up-regulation of *MT1-MMP* is involved in the mechanism by which HCV induces invadopodia matrix degradation, we knocked down *MT1-MMP* in HCV-infected and non-infected HCC cells and measured their ability to degrade fluorescently-labeled gelatin as above (Figure 3C, left). As demonstrated in Figure 3C (right), knockdown of *MT1-MMP* in HCV-infected cells significantly reduced their ability to degrade the extracellular matrix. This result validated the involvement of *MT1-MMP* in the mechanism by which HCV regulated invadopodia matrix degradation in infected cells.

Collectively, these results demonstrated that HCV infection up-regulated invadopodia formation and function that consequently enhanced matrix degradation and cell invasion.

### 3.4. HCV Infection Induces Activation of Receptor and Non-Receptor Tyrosine Kinases

Transformation of invadopodium precursors into mature, matrix-degrading invadopodia is governed by activation of tyrosine kinases, such as EGFR, Pyk2, Src, and Arg, which control specific signaling pathways leading to functional activation of invadopodia [41,49]. Thus, we speculated that higher expression of invadopodia-related genes is not sufficient to activate invadopodia, and activation of their function by tyrosine kinases is also required. To investigate the role of tyrosine kinases in the activation of invadopodia following HCV infection of HCC cells, we performed a functional Kinome screen using the tyrosine kinase PamChip array. Using the array, we measured the differential activation of tyrosine kinases in cell lysates from HCV-infected (approximately 100% infection, prepared as described in the methods) and non-infected cells (Figure 4A,B). The heatmap presented in Figure 4A points for an altered pattern of tyrosine kinases that are activated upon infection, compared to non-infected cells. Among the top-scored tyrosine kinases, we identified EGFR, which is known to regulate invadopodia maturation and activation (Figure 4B). Moreover, the physical and functional association network of the top affected kinases revealed that EGFR was also the most interconnected tyrosine kinase (Figure 4C) [39].

The intersection of the list of activated tyrosine kinases with invasion- or invadopodia-related gene lists that were generated by literature mining revealed several receptor tyrosine kinases (RTKs) as well as non-receptor tyrosine kinases (NRTKs) that are activated in HCC cells following HCV infection (Appendix A). Among the NRTKs involved in invadopodia activation, we identified EPHB2, a member of the ephrin receptor family. Multiple other members of the ephrin family were among the top-scored tyrosine kinases activated following HCV infection (Appendix A). Among the top-scored RTKs, we observed significant activation of multiple EGFR family members, such as EGFR, ERBB2, ERBB3, and ERBB4, following HCV infection (Appendix A). Indeed, a previous report demonstrated that the functions of EGFR and EPHA2 play essential roles in HCV entry [51].

Tyrosine kinases can be activated by either overexpression, which increases the probability for dimer formation, or by ligand binding. To investigate which of these two mechanisms leads to the activation of tyrosine kinases in our system, we intersected the list of activated kinases from our PamChip array with the list of up-regulated genes from RNA-seq (Figure 4D and Appendix A). Interestingly, we found several tyrosine kinase receptors that were up-regulated in terms of activity (PamGene, the left group) and bound to their cognate ligands, which were up-regulated by gene expression (RNA-seq, the right group). Among these, we detected members of the EGFR family and their ligands, specifically HB-EGF, NRG1, and BTC. These ligands activated EGFR, as well as ABL1, EPHA6, and SRC that were phosphorylated and activated by EGFR. An additional overlapped group included non-receptor tyrosine kinases and two receptors, EPHA6 and INSR, which were both up-regulated by gene expression and showed increased kinase activity. These data supported the hypothesis that HCV enhanced tyrosine kinase activation in infected cells by increasing receptor- and non-receptor tyrosine kinase gene expression, as well as by up-regulating the expression of ligands that might induce autocrine activation of their respective receptor tyrosine kinases.

### 3.5. HCV Promotes Invadopodium Precursor Formation and Activation via an NS3/4A-TC-PTP-EGFR Pathway

It was previously shown that HCV influences the activation of the EGFR signaling pathway through an NS3/4A-mediated cleavage and consequent down-regulation of TC-PTP. This down-regulation of TC-PTP results in an enhancement of EGFR-induced signal transduction in the infected host cell, which is essential for the maintenance of efficient viral replication [52]. Because our kinase activation assay revealed EGFR as the most activated and inter-connected tyrosine kinase in HCV-infected cells, we hypothesized that a similar EGFR activation mechanism was used by HCV for inducing assembly and activation of invadopodia.

Previous reports demonstrated a link between enhanced motility and invasiveness of hepatocytes and HCC cells to overexpression of the HCV oncogenic proteins, mainly NS3/4A and Core [22,24,29,30,31,32]. To investigate whether one of these viral proteins controls the invadopodium precursor assembly, we ectopically expressed the Core protein or NS3/4A protease in HCC cells (Figure 5A). These cells were plated on gelatin matrix and labeled for the invadopodium precursor markers—actin and cortactin. As demonstrated in Figure 5B, NS3/4A-transfected cells, but not Core-transfected or control cells, showed a significant increase in invadopodium precursors. To evaluate the effect of NS3/4A and Core on the regulation of MMP-mediated extracellular matrix degradation by mature invadopodia, cells expressing either viral protein were plated on a fluorescently labeled gelatin matrix. Quantification of the degradation area by the cells revealed a significant increase in the ability of NS3/4A expressing cells, but not Core-transfected or control cells to degrade the matrix (Figure 5C). Collectively, these data suggested that invadopodium precursor assembly and activation were regulated by the HCV protease NS3/4A.

To determine whether NS3/4A controls invadopodium precursor assembly and activation via the EGFR signaling pathway, we repeated the invadopodium precursor assembly and matrix degradation experiments by ectopically expressing the NS3/4A protease in HCC cells in the presence of the EGFR kinase inhibitor erlotinib. As demonstrated in Figure 5D,E, and in agreement with our previous data above, overexpression of the viral protease in HCC cells significantly increased invadopodium precursor assembly and the consequent matrix degradation in these cells compared to control-transfected cells. Importantly, invadopodium precursor formation and matrix degradation were significantly reduced in NS3/4A overexpressing cells following erlotinib treatment, suggesting that invadopodium assembly and maturation is regulated by the viral protease NS3/4A in an EGFR-mediated mechanism.

To test whether regulation of EGFR activation by the viral protease NS3/4A is mediated via down-regulation of the tyrosine phosphatase TC-PTP, we repeated the invadopodium precursor assembly and matrix degradation experiments using HCV-infected and non-infected cells in which TC-PTP was ectopically expressed by transfection (Figure 5A). Endogenous TC-PTP was cleaved in both HCV-infected cells and NS3/4A-transfected cells, as previously demonstrated [52] (Figure 5F). However, ectopically expressed TC-PTP was not completely cleaved, and overexpression resulted in residual intact protein (Figure 5G). As demonstrated in Figure 5H,I, and confirming our previous results, infection of HCC cells with HCV significantly increased precursor formation as well as matrix degradation, while overexpression of TC-PTP in these cells resulted in a significant reduction in precursor numbers and gelatin degradation. Collectively, these data suggested that HCV induced an NS3/4A-mediated down-regulation of TC-PTP and consequent enhancement of EGFR activation and downstream signaling, leading to invadopodium precursor assembly and activation in HCV-infected cells.

### 3.6. HCV Infection Enhances Intrahepatic and Extrahepatic Dissemination of HCC Cells

The enhanced motility or invasion properties of HCV-infected cancer cells we observed may result in gaining higher intra and extrahepatic cell dissemination in vivo. However, the effect of the infectious HCV on the dynamics of the in vivo invasiveness and metastatic dissemination of HCC tumor cells has not been explored. To investigate the effect of HCV infection on the growth of HCC cells that are implanted within their natural microenvironment, we used an orthotopic xenograft model. Huh7.5 HCC cells infected with HCV (approximately 100% infection, prepared as described in the methods) or non-infected cells were injected into the left lobe of mice livers. Following four weeks post-injection, the HCC cells were detected by injection of fluorescently labeled 2-Deoxy-D-glucose (2DG), which is a marker for tumor cells that have high glucose uptake. As demonstrated in Figure 6A, the intensity of the fluorescent signal detected in the liver was higher in mice injected with HCV-infected cells compared to mice injected with non-infected cells, but this difference was not significant. These results suggested that HCV infection did not significantly affect tumor growth in vivo.

We next aimed to evaluate whether although HCV infection does not affect the growth of HCC cells, it influences their spread within the liver. Therefore, to investigate specifically the dynamics of cancer cell spread within the liver and its metastatic dissemination to other organs, we applied a highly sensitive assay that is based on qRT-PCR for the parallel amplification of unique, species-specific conserved, and non-transcribed sequences in the mouse and human genomes using human or mouse-specific primers (Appendix A) [45]. Equal amounts of HCV-infected or non-infected HCC cells were injected into the left lobe of the liver of immune-deficient mice (seven mice for each test group, a total of 28 mice). At two and four weeks following injection, all mice that survived the surgery (at least five mice per test group) were sacrificed, and livers and lungs were harvested, and the human tumor cells within the injected left liver lobe, the remaining liver lobes (right lobes), and the lungs were quantified (Figure 6B). No change in the percentage of non-infected human cancer cells was observed between two and four weeks post-injection in the left lobe of the liver, where cells were originally injected. In contrast, a significant reduction in the percent of HCV-infected cells that remained in the left lobe of the liver was observed in both two and four weeks following injection (Figure 6C, left). Examination of the relative amount of human cancer cells in the right lobes of the liver revealed an opposite phenotype. While no significant change was observed in the percent of non-infected cells between two and four weeks of injection, a significantly higher percent of HCV-infected cells was observed at four weeks compared to two weeks and at four weeks compared to non-infected cells (Figure 6C, middle). Similar results were observed for the percentage of HCV-infected versus non-infected human cancer cells in mouse lungs (Figure 6C, right). These data suggested that infection with HCV enhanced the invasion properties of HCC cells that resulted in cells invading from the left lobes to the right lobes and the lungs, at a higher rate compared to non-infected cells. Collectively, the data demonstrated that HCV infection did not increase tumor growth, but significantly increased the cells’ in vivo intrahepatic and extrahepatic dissemination capabilities.

To validate these results, specimens from the left and right lobes of the liver were sectioned and stained for human HLA. Representative images that are presented in Figure 6D demonstrate the presence of human cells in the mouse tissues. Quantification of the stained area from at least 10 random areas from each slide (at least three mice for each left/right lobes) demonstrated higher staining of human cells in mice injected with HCV-infected cells in the left lobe, and lower staining in the right lobes, compared to staining of mice injected with non-infected cells. These results confirmed the data obtained with the qRT-PCR assay (Figure 6E).

We next sought to determine whether the invadopodia-related gene *CTTN*, encoding for cortactin, is also expressed in vivo during the initial process of intrahepatic invasion. qRT-PCR using primers that specifically recognize the human invadopodia-associated gene *CTTN* (Appendix A) was performed for left and the right lobes of the liver samples at two weeks post-injection. As demonstrated in Appendix A, higher mRNA expression of *CTTN* was observed in the left lobe of the liver of mice injected with HCV-infected cells compared to non-infected cells. This result suggested that invadopodia were formed in HCV infected cells in the left lobe (the site of injection), which consequently enhanced invasion of infected cells to the right lobes compared to non-infected cells in which cortactin expression was not detected.

### 3.7. Invadopodia-Related Gene Expression Correlates with Aggressiveness of HCC and with Poor Patient Prognosis

The extremely poor prognosis of HCC results from high recurrence rate after surgery or from intrahepatic metastases that develop due to the dissemination of tumor cells, which either locally invade or distantly spread via the portal vein into other parts of the liver. Our data suggested that HCV infection significantly increased invasiveness and metastatic spread of virus-infected tumor cells due to the enhancement of invadopodium precursor formation and activation. To evaluate the implications of these results to human disease, we investigated the correlation between invadopodia-associated gene expression to the overall survival (OS; the length of time after tumor resection in which the patient is still alive) and disease-free survival (DFS; the length of time after tumor resection without any signs or symptoms of the cancer) of HCC patients. A significant correlation was found between higher expression of invadopodia-related genes to decreased overall survival and/or disease-free survival of HCC patients with a background of HCV infection, while no similar correlation was observed in the overall population of HCC patients. This suggested that viral-induced overexpression of invadopodia-associated genes might promote invasion and disease progression in HCV-related HCC patients (Figure 7). Altogether, this analysis confirmed the significance of invasiveness in HCC progression and further validated our data, suggesting an association between HCV infection, increased invadopodia-mediated cancer invasiveness and consequent cancer aggressiveness, and poor patient prognosis.

## 4. Discussion

A major milestone in early cancer research was the discovery of oncogenic human viruses, such as the Rous sarcoma virus, which carried the viral tyrosine kinase Src [53]. Years later, the transformation of fibroblasts by infection with the same virus was shown to lead to the appearance of cellular feet-like structures that degrade the extracellular matrix, termed as invadopodia [54]. Here, we showed for the first time that HCV led to the regulation of invadopodia formation and activation and, consequently, to the invasive phenotype of infected HCC cells. To our knowledge, this is the first example of a viral-induced invasion and metastatic dissemination of cancer cells by directly regulating invadopodia formation and activation. Moreover, this is the first demonstration of in vivo intra and extrahepatic invasiveness and metastatic spread of HCC cells expressing infectious HCV when implanted within their natural microenvironment in the liver.

Based on the data presented herein, we suggested a model by which HCV controls invadopodium precursor formation and activation by combining enhancement of gene expression with stimulation of kinase signaling (Figure 8). According to our model, once inside the cell, HCV induces sustained phosphorylation and activation of EGFR by the protease function of its NS3/4A protein, which cleaves TC-PTP and inactivates it. As a result, EGFR remains tyrosine phosphorylated and activated for a prolonged time. As a result, the activated EGFR, which is a key regulator of invadopodia formation and activation, as we previously showed [41], transmits signals to downstream non-receptor tyrosine kinases and other invadopodia proteins, which leads to the maturation and activation of invadopodium precursors in the host cell. HCV also regulates the transcription of invadopodia-associated genes. Among the transcribed genes is *MT1-MMP*, which leads to enhancement of extracellular matrix degradation, and ligands for host cell receptors, such as *HB-EGF* and *BTC* for EGFR, leading to sustained activation of the receptors in an autocrine loop. The combination of these HCV-induced events leads to invadopodia-dependent enhancement of tumor cell invasiveness and consequent intra and extrahepatic cancer cell dissemination. Our patient gene expression database analysis revealed a correlation between invadopodia-associated gene expression signature and poor patient survival, suggesting a link between HCV-mediated invadopodia and HCC aggressiveness.

Since the viral life cycle of HCV relies completely on the host cell infrastructure, and to permit efficient virus internalization and persistent replication of the viral genome without affecting host cell viability and survival, HCV has evolved mechanisms for using and modifying host cellular signaling pathways involved in proliferation and cell survival. Among these signaling pathways are the EGFR pathway, PI3K/Akt pathway, and cytoskeletal pathways involved in endocytosis and vesicular transport [55]. Here, we suggested that the HCV-induced modulation of these host cytoskeletal signaling pathways, which are important for its life cycle, also led to invadopodia formation and activation and, consequently, cancer cell invasiveness. It is reasonable to speculate that by enhancing the spread of HCC cells to new regions away from the primary tumor, HCV gains a new environment in which the cancer cells can survive and thrive without the need of competing on nutrients and oxygen resources with other tumor cells, therefore enabling virus survival and its further amplification. Another interesting hypothesis is that HCV enhances cancer cell motility and invasion into the new environment in the liver to infect new cells and increase overall virus replication and production. This hypothesis raises the question of whether HCV infection increases the invasiveness of healthy hepatocytes or only of tumor cells.

In principle, to initiate the metastatic process, cells must first acquire motility and invasive abilities, allowing their transport to distant sites or organs. At secondary sites, carcinoma cells can form a new carcinoma (metastasis) through a mesenchymal-epithelial transition (MET), in which cells acquire the ability to proliferate. As previously shown by Akkari et al. [19], expression of HCV proteins in primary hepatic precursors and immortalized hepatocyte cell lines induced EMT and cell migration and invasion in vitro. However, in a mice xenograft model, where the cells expressing viral proteins were injected subcutaneously, no tumors were observed. Importantly, the addition of Ras following injection of HCV proteins-expressing cells induced extensive tumor formation and metastasis in the lungs. These results indicate that, indeed, infected primary hepatocytes do acquire migratory and invasive properties, supporting our hypothesis that the properties of invasion of the infected cells might provide an advantage for HCV for its survival and spread, also at initial stages of infection. However, in primary hepatocytes, the presence of HCV may not be sufficient for tumor formation and metastasis, and additional factors that are produced by tumor cells, such as Ras, are required for the enhanced aggressiveness and metastasis formation in vivo. Collectively, this suggests that the metastatic process in HCV-infected cells occurs in the late-stage following the progression to cancer.

Recent data showed conflicting results regarding the benefit of DAAs on HCC incidence, and several reports even raised concerns about an increased risk of HCC occurrence and recurrence after DAAs therapy, although the finding was not replicated in a subsequent study [56,57,58,59,60]. The following studies reported not only a higher recurrence rate of HCC among the DAAs-treated patients but also a more aggressive pattern of recurrence, rapid development, and more vascular invasion following SVR [61]. Moreover, increase de novo HCC incidence despite achieving SVR was reported [62,63]. In contrast, several other reports concluded that DAAs are not associated with a significant risk of HCC as compared to interferon-based treatment [64,65]. Nonetheless, these studies demonstrated that while there was a relative risk reduction in HCC, the absolute risk of HCC persisted in those patients who had DAAs-induced SVR. Considering the expected growing number of DAAs-treated HCV patients in the near future, these observations highlight the need for finding new therapeutic options, also specifically for HCV-induced metastatic cancer that may be different from therapies targeting solid HCC tumors. We recently revealed an epigenetic memory mechanism that preserves the HCV-specific oncogenic effects on the host cells, mainly the expression of genes associated with cytoskeleton remodeling, also after the virus is cured in the cells by DAAs [35]. Moreover, erlotinib reverted this persistent epigenetic signature. Although our observations should be evaluated in clinical settings, they may suggest that drugs that target the HCV-mediated mechanism of metastasis, such as erlotinib, may be useful in combination with the standard drugs for HCC to decrease cancer recurrence and dissemination, also after SVR by DAAs. Moreover, we found a unique invadopodia gene expression signature that characterizes HCV-mediated HCC tumors, that may be used as a prognostic tool to predict aggressiveness and invasiveness of HCV-mediated HCC pre and post SVR.

## Figures and Tables

**Figure 1 cells-08-01395-f001:**
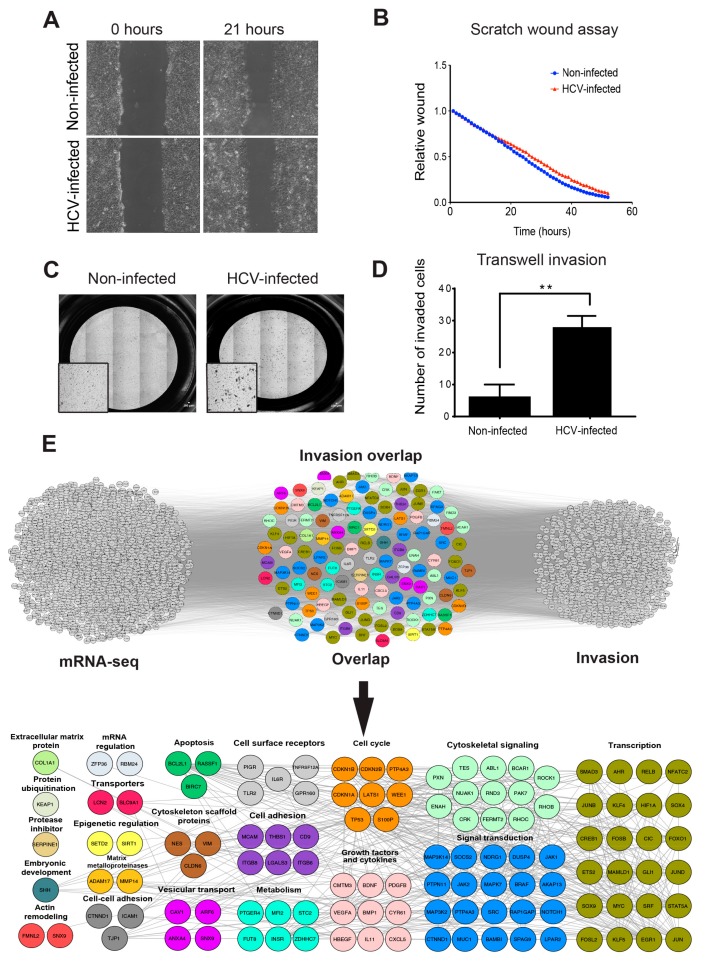
HCC (hepatocellular carcinoma) cell invasion, but not migration, is enhanced by HCV (hepatitis C virus) infection. (**A**,**B**) Non-infected or HCV-infected cells were plated and allowed to form a monolayer. At 24 h following plating, mitomycin was added to the monolayer for 2 h in order to inhibit cell proliferation, and a scratch was performed. Cells were exposed to live-imaging for 53 h during which wound closure was measured. Shown are representative images at time 0 and 21 h following wounding (**A**) and quantification of wound closure over 53 h (**B**). *n* = 6 fields from three independent experiments. (**C**,**D**) Non-infected or HCV-infected cells were plated on the upper chamber of Matrigel-coated Transwells and allowed to invade for 24 h. Filters were stained with crystal violet (**C**), and the cells that invaded into the lower side of the filters were counted. *n* = 10 fields from three independent experiments (**D**). * *p* ≤ 0.01, Student’s *t*-test. (**E**) Top: overlap of overexpressed genes as measured by RNA sequencing (left) and invasion-associated gene list obtained by literature mining (right). The lists of genes were overlayed using Cytoscape, and the combined list of common nodes was used to build a physical and functional association network in STRING (middle). Bottom: close-up view of the overlapped gene network. Genes were classified based on GeneCards and UniProt identified function.

**Figure 2 cells-08-01395-f002:**
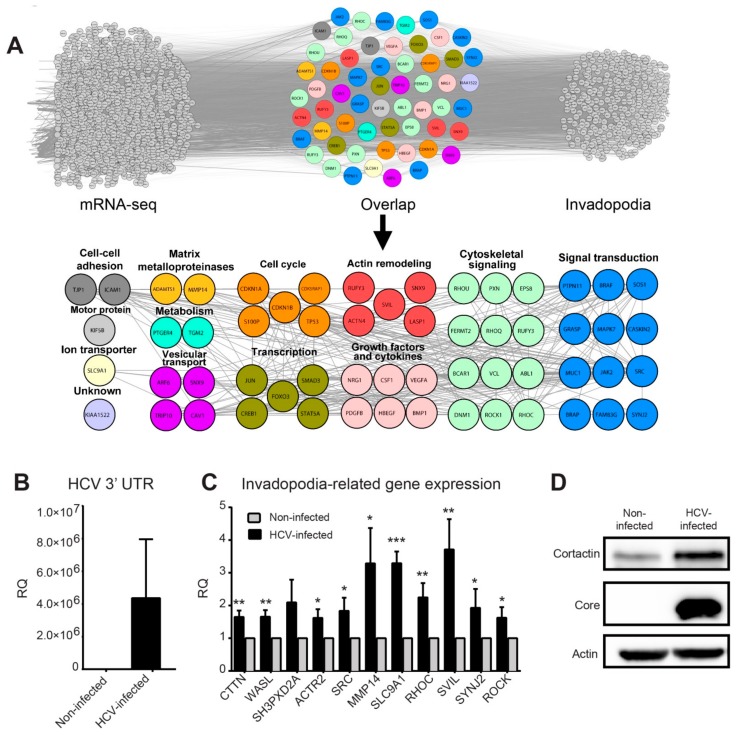
HCV infection induces the expression of invadopodia-associated genes. (**A**) Top: overlap of overexpressed genes as measured by RNA sequencing (left) and invadopodia-associated gene list obtained by literature mining (right). The lists of genes were overlayed using Cytoscape, and the combined list of common nodes was used to build a physical and functional association network in STRING (middle). Bottom: close-up view of the overlapped gene network. Genes were classified based on GeneCards and UniProt identified function. (**B**,**C**) qRT-PCR evaluation of HCV infection efficiency (**B**) and selected invadopodia-associated genes from non-infected and HCV-infected Huh7.5 HCC cells (**C**). * *p* < 0.05; ** *p* < 0.01; *** *p* < 0.001, Student’s *t*-test. The tested genes include *CTTN* (cortactin), *WASL* (N-WASP), *SH3PXD2A* (TKS5), *ACTR2* (ARP2), *SRC*, *MMP14* (MT1-MMP), *SLC9A1*, *RHOC*, *SVIL*, *SYNJ2*, and *ROCK*. Values were presented as relative quantity (RQ) compared to non-infected cells. Shown are results from three independent experiments. (**D**) Cell lysates from infected and non-infected cells were analyzed by western blot using antibodies for cortactin (Abcam), core (Thermo Fisher Scientific), and β−actin (Sigma).

**Figure 3 cells-08-01395-f003:**
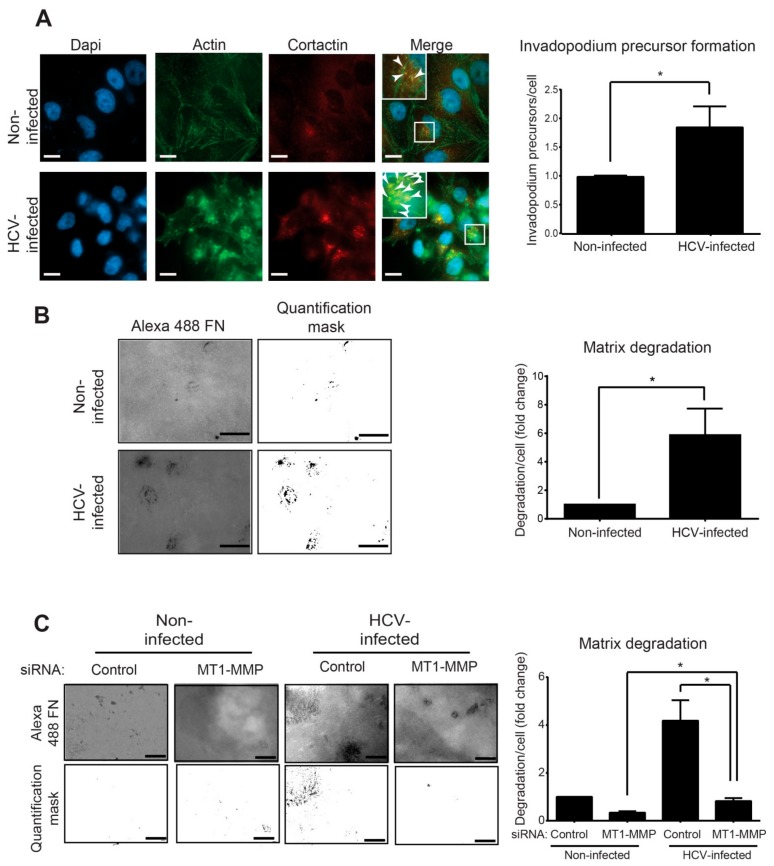
Infection with HCV enhances invadopodium precursor formation and activation. (**A**) Left: HCV-infected and non-infected Huh7.5 cells were plated on unlabeled gelatin, fixed, and immunostained for actin (green) and cortactin (red). Boxed regions and insets depict localization of actin dots and cortactin as markers of invadopodium precursors. Bar, 5 μm. Right: Quantification of invadopodium precursors per cell in non-infected and HCV-infected cells. *n* = 40 cells per group from three independent experiments. (**B**) Left: HCV-infected and non-infected Huh7.5 cells were plated on Alexa 488 gelatin and allowed to degrade for 72 h. Shown are representative images (left panel) and quantification masks (right panel) of degradation areas. Bar, 5 μm. Right: Quantification of matrix degradation by non-infected and HCV-infected cells. *n* = 10 fields per group from three independent experiments. (**C**) Left: HCV-infected and non-infected Huh7.5 cells were transfected with control siRNA or with *MT1-MMP* siRNA for four hours. Cells were then plated on Alexa 488 gelatin and allowed to degrade for 72 h. Shown are representative images (left panel) and quantification masks (right panel) of degradation areas. Bar, 5 μm. Right: Quantification of matrix degradation by non-infected and HCV-infected cells expressing either control or *MT1-MMP* siRNA. *n* = 10 fields per group from three independent experiments. * *p* < 0.05, Student’s *t*-test.

**Figure 4 cells-08-01395-f004:**
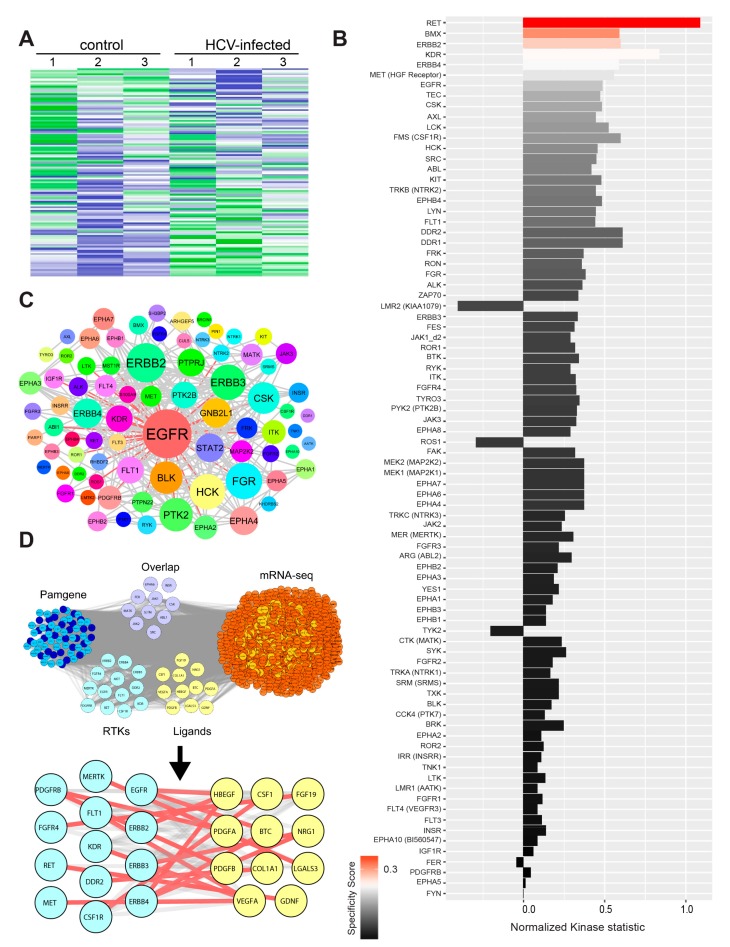
HCV infection induces activation of receptor and non-receptor tyrosine kinases. (**A**) Upstream kinase analysis of HCV-infected and non-infected cell lysates. The Z-score of the activated tyrosine kinases in non-infected and HCV-infected Huh7.5 was presented in a heatmap generated by using the Bionavigator software. Three biological replicates are shown for each. (**B**) The plot shows tyrosine kinases ranked according to their score. The difference between non-infected and HCV-infected samples was analyzed using sets of peptides that are predicted to be substrates of the respective kinases. The length of bars represents the kinase statistics, a measure for the change in activity between the two groups (significance). A positive value of the kinase statistics indicates that the associated kinase activity was higher in the HCV-infected group. The color of bars indicates the specificity of the kinase set. (**C**) Physical and functional association network of the top putatively affected kinases using Cytoscape. Node sizes were proportionally mapped to the connectivity degree and demonstrate EGFR to be the most prominent molecular hub. Edges connecting EGFR with its neighbors are shown in red. (**D**) Overlap of activated tyrosine kinases (Pamgene) with overexpressed RNA-seq genes (purple) and depiction of the activated receptor tyrosine kinases (cyan) and their respective overexpressed RNA-seq extracellular ligands (yellow). Overexpressed genes are colored in light red, and nodes interacting with Pamgene kinases in orange. Activated Pamgene tyrosine kinases were divided into receptors (cyan) and non-receptors (blue). Receptor-ligand complex interactions were manually curated using GeneCards and UniProt and are represented in the close-up view as red lines. The lists of genes were overlayed using Cytoscape, and the combined list of common nodes was used to build a physical and functional association network in STRING.

**Figure 5 cells-08-01395-f005:**
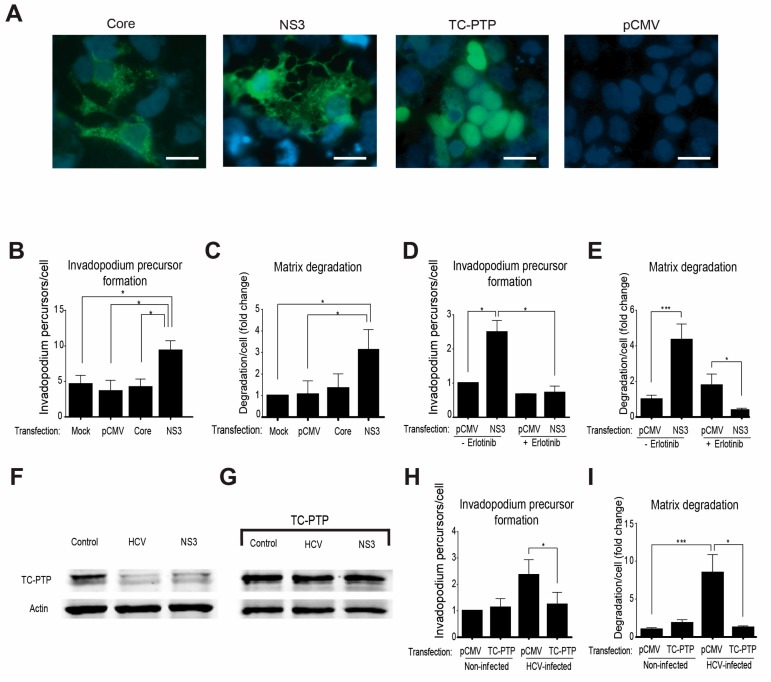
HCV promotes invadopodium precursor formation via down-regulation of T-cell protein tyrosine phosphatase (TC-PTP) and activation of epidermal growth factor receptor (EGFR). (**A**) Huh7.5 cells transfected with plasmids containing FLAG-tagged HCV-Core, FLAG-tagged HCV-NS3/4A, FLAG-tagged TC-PTP, or empty vector (pCMV) as control were immunostained with anti-core (for the core) positive serum from HCV-infected patient (for NS3) or anti-TC-PTP. Bar, 5 μm. (**B**) Transfected cells were plated on unlabeled gelatin, fixed, and labeled for actin and cortactin. Co-localization of actin and cortactin dots as markers of invadopodium precursors was counted. Presented is the quantification of invadopodium precursors per cell. *n* = 20 cells per group from three independent experiments. (**C**) Huh7.5 cells were transfected as above, plated on Alexa 488 gelatin, and allowed to degrade for 72 h. Presented is the quantification of matrix degradation by control and transfected cells. *n* = 10 fields per group from three independent experiments. (**D**) Huh7.5 cells were plated on unlabeled gelatin for 24 h and then transfected with plasmids containing FLAG-tagged HCV-NS3/4A or empty vector (pCMV) as a control. Erlotinib (1 μM) was added six hours following transfection for additional 24 h. Cells were then fixed and labeled for actin and cortactin as invadopodia markers. Invadopodium precursor formation (*n* = 20 cells per group from three independent experiments) was quantified. (**E**) Matrix degradation of Huh7.5 cells transfected and treated with Erlotinib (*n* = 10 fields per group from three independent experiments) was quantified. (**F**) Cell lysates from non-infected, HCV-infected, and NS3-transfected cells were analyzed by western blot using antibodies for TC-PTP (Abcam) and β actin (Sigma). (**G**) Cell lysates from non-infected, HCV-infected, and NS3-transfected cells, all transfected with FLAG-tagged TC-PTP, were analyzed by western blot using antibodies for TC-PTP (Abcam) and β actin (Sigma). (**H**) HCV-infected or non-infected Huh7.5 cells were transfected with a plasmid containing FLAG-tagged TC-PTP or with empty vector (pCMV) as control, plated on unlabeled gelatin for 24 h, fixed and labeled for actin and cortactin as invadopodia markers. Invadopodium precursor formation in cells that showed a positive signal in anti-FLAG staining was quantified. *n* = 15 cells per group from two independent experiments. (**I**) Quantification of matrix degradation of Huh7.5 cells transfected as above is presented. *n* = 10 fields per group from three independent experiments. * *p* < 0.05; *** *p* < 0.001, Student’s *t*-test.

**Figure 6 cells-08-01395-f006:**
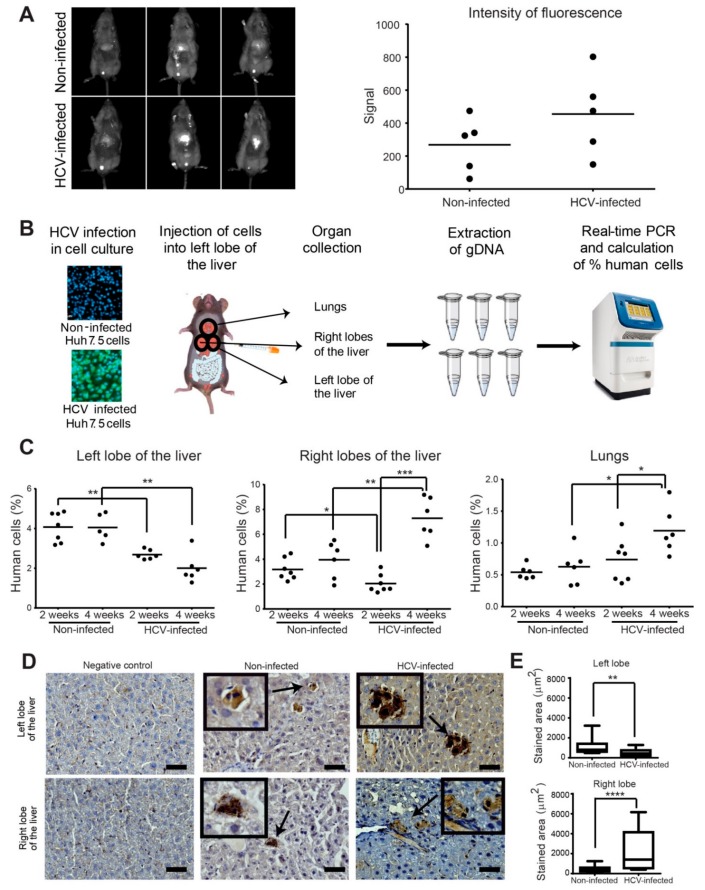
HCV enhances intrahepatic and extrahepatic dissemination of infected HCC cells. (**A**) In vivo fluorescence images (left) and quantification (right) of overall tumor cell load in the liver of mice injected with non-infected (top) or HCV-infected (bottom) Huh7.5 cells. Images were acquired four weeks post tumor cell injection. *n* = 5 mice per group. (**B**) HCV-infected or non-infected Huh7.5 control cells were injected into the left lobe of the liver of NSG male mice. Two and four weeks following injection, the left lobe, right lobes, and lungs of the mice were collected and dissociated, genomic DNA was extracted, and human/mouse DNA content was evaluated by quantitative real-time PCR. (**C**) Quantification of the percentage of human cells in the left lobe (left graph), right lobes (middle graph), and lungs (right graph) of mice injected with non-infected or HCV-infected Huh7.5 cells. Mouse tissues were collected at two and four weeks post-injection. *n* = 6–7 mice per group. (**D**) Representative immunohistochemistry staining for H&E and human HLA of the negative control (left panels), non-infected (middle panels), and HCV-infected (right panels) human HCC cells in the left lobe of the liver (top) and the lungs (bottom) of mice. Bar, 5 μm. (**E**) Quantification of the stained area from at least 10 random areas from each slide (at least three mice for each left/right lobes). * *p* < 0.05; ** *p* < 0.01; *** *p* < 0.001; **** *p* < 0.0001, Student’s *t*-test.

**Figure 7 cells-08-01395-f007:**
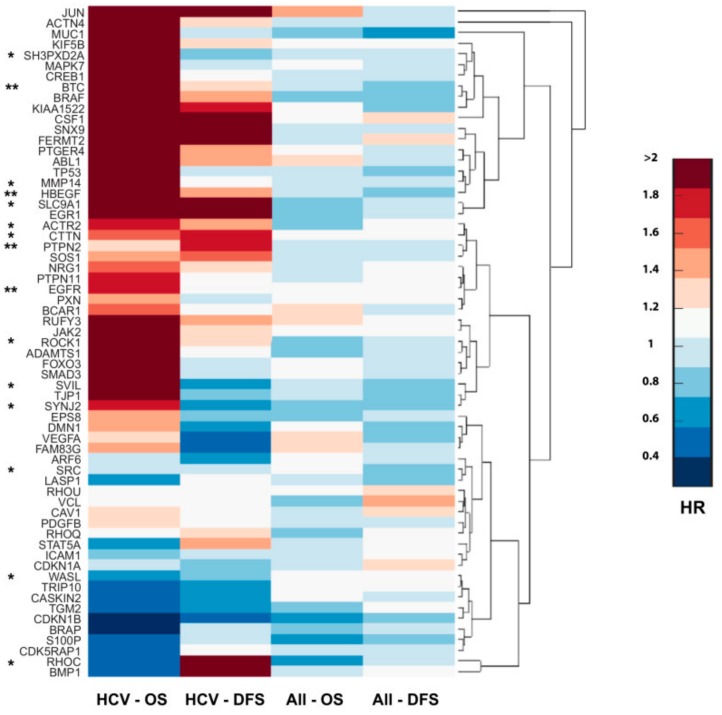
Invadopodia-related gene expression correlates with the aggressiveness of HCV-mediated HCC and poor patient prognosis. Cluster gram of overexpressed invadopodia-associated genes by hazard ratio (HR). Columns represent HR values of OS (overall survival) and DFS (disease-free survival) in HCC patients (*n* = 377) and HCV-related HCC (*n* = 47). Genes are clustered in groups where the upper clusters are associated with the worst survival in HCV-related HCC. qRT-PCR validated genes are marked by an asterisk (*), and *EGFR*, TC-PTP (*PTPN2*), *HB-EGF*, and *BTC* are marked by two asterisks (**).

**Figure 8 cells-08-01395-f008:**
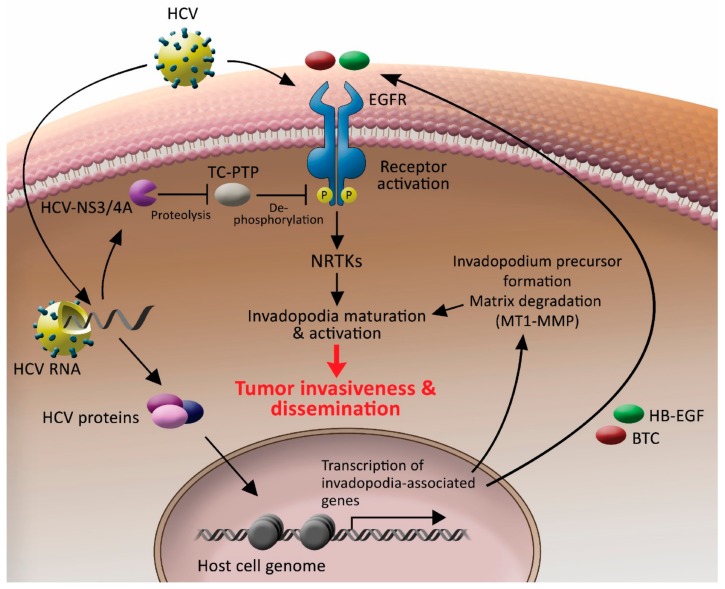
HCV controls invadopodium precursor formation and activation by combining enhancement of gene expression with stimulation of kinase signaling. HCV enters the host cell by using EGFR as a co-receptor. Once the virus is inside the cell, it sustains the activation of the receptor by activating the NS3/4A-TC-PTP axis and sustaining receptor tyrosine phosphorylation. Activated EGFR transmits signals to downstream non-receptor tyrosine kinases and other invadopodia proteins, which leads to the maturation and activation of invadopodium precursors in the host cell. HCV also misregulates chromatin organization and, consequently, transcription of host invadopodia-associated genes. Among the transcribed genes is *MT1-MMP*, which leads to enhancement of extracellular matrix degradation, and *HB-EGF* and *BTC* ligands for EGFR, leading to sustained activation of the receptor in an autocrine loop. A combination of these HCV-induced events leads to invadopodia-dependent tumor cell invasiveness and, consequently, intra and extrahepatic cancer cell dissemination.

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
