# Peer review of "Hepatitis C Virus Enhances the Invasiveness of Hepatocellular Carcinoma via EGFR-Mediated Invadopodia Formation and Activation"

_cells, 2019, doi:10.3390/cells8111395_

Round 1
Reviewer 1 Report
This is a well-written manuscript with well-organized design. However, I have a few concerns:
Major points
Tks5, a major moderator of invadopodia formation, is not included in the list of the differentially expressed genes. Check TKS5 expression in control and HCV infected cells. If difference is noted, additional knock-down experiment may support the authors’ hypothesis more convincingly. To my understanding, invadopodia precursor formation is an early stage phenomenon, whereas MT1-MMP acts in the later stage of invadopodium enlongation / ECM degradation. What is the driver of increased invadopodia precursor formation in HCV infected cells? Also provide the invadopodium precursor formation data in control- / siRNA against MT1-MMP-treated cells Provide the p value for Fig 6A and explanations for the asterisks in Fig 6B.
Minor points
In Supplementary Figure 1, the scale seems different between the control and HCV infected cells, although the scale bars showed same length.
Author Response
Point 1: Tks5, a major moderator of invadopodia formation, is not included in the list of the differentially expressed genes. Check TKS5 expression in control and HCV infected cells. If difference is noted, additional knock-down experiment may support the authors’ hypothesis more convincingly.
Response 1: Indeed, TKS5 is essential for invadopodium precursor formation. Therefore, although change of expression of the gene for TKS5, SH3PXD2A, was not detected by RNA-seq (Figure 2A), still we performed qRT-PCR for this gene that may be a more sensitive method to measure changes in gene expression comparing to RNA-seq (Figure 2C). Indeed, an increase in expression of the gene SH3PXD2A was observed in HCV-infected vs. non-infected cells in qRT-PCR assay, however, the change was not significant (Figure 2C). Following these results, we feel that performing knockdown of TKS5 will not be an appropriate assay, since the enhancement of invadopodia formation in HCV infected cells seems not to be mediated by increased levels of TKS5. The results shown in Figures 1 and 2 points for a complex process that leads to the enhancement of invadopodia formation in HCV infected cells that requires the orchestrated increased expression level of multiple invadopodia and invasion related genes rather than mediated by a change in a single gene expression.
We understand that the nomenclature of gene names was confusing and in the revised manuscript we added more explanations in Figure 2 legend and in line 377-383. We appreciate the reviewer’s note on this specific issue, that led us to clarify the related text.
Point 2: To my understanding, invadopodia precursor formation is an early stage phenomenon, whereas MT1-MMP acts in the later stage of invadopodium elongation/ECM degradation. What is the driver of increased invadopodia precursor formation in HCV infected cells?
Response 2: Indeed, as noted by the reviewer, the invadopodium life cycle contains several stages which start in the precursor formation and ends in its maturation which leads to the activation of MMPs and ECM degradation. Our model, presented in Figure 8, suggests that infection with HCV affects both invadopodium precursor formation as well as invadopodia maturation and activation, by inducing invadopodial core gene transcription and activating receptor and non-receptor tyrosine kinases which are known for activating invadopodia. More specifically, invadopodium precursors are composed of core proteins such as cortactin and Src, which are shown to increase in their expression in HCV-infected cells, as demonstrated by our RNA-seq results as well as and by the qRT-PCR in Figure 2C. Moreover, the constitutive activation of EGFR, which is known to induce invadopodia formation and activation as we previously showed (Mader et al, 2011), by HCV NS3 protein leads to the increased invadopodia precursor formation in HCV infected cells.
Following the reviewer’s note, we have modified the text in lines 677-693 which explains the model presented in Figure 8 for the drivers of increased invadopodia precursor formation and invasion in HCV infected cells.
Point 3: Also provide the invadopodium precursor formation data in control- / siRNA against MT1-MMP-treated cells.
Response 3: As noted above by the reviewer, MT1-MMP is involved in the late steps of maturation and activation of invadopodia and not in the invadopodium precursor formation stage. Therefore, a knockdown of MT1-MMP will not affect the number of invadopodia precursors.
Point 4: Provide the p value for Fig 6A and explanations for the asterisks in Fig 6B.
Response 4: The differences in Figure 6A are not significant and therefore P-value is not provided. This point is explained in the text line 578: “As demonstrated in Figure 6A, the intensity of the fluorescent signal detected in the liver was higher in mice injected with HCV-infected cells compared to mice injected with non-infected cells, but this difference was not significant. These results suggest that HCV infection do not significantly affects tumor growth in vivo. “
Regarding the asterisks in Figure 6B, we have added an explanation in the figure legend of the revised manuscript on page 19, and to all other figure legends. We thank the reviewer for this constructive comment.
Point 5: In Supplementary Figure 1, the scale seems different between the control and HCV infected cells, although the scale bars showed same length.
Response 5: We appreciate the reviewer comment, and acknowledge that size of nuclei may appear different in infected compared to non-infected cells, however we rechecked the scale, and it is correct. The different appearance of the nuclei in infected cell may be due to changes that occur in the nuclei following infection, as we have recently reported (Perez et al, 2019).
References
Mader CC, Oser M, Magalhaes MA, Bravo-Cordero JJ, Condeelis J, Koleske AJ, Gil-Henn H (2011) An EGFR-Src-Arg-cortactin pathway mediates functional maturation of invadopodia and breast cancer cell invasion. Cancer Res 71: 1730-1741
Perez S, Gevor M, Davidovich A, Kaspi A, Yamin K, Domovich T, Meirson T, Matityahu A, Brody Y, Stemmer SM, El-Osta A, Haviv I, Onn I, Gal-Tanamy M (2019) Dysregulation of the cohesin subunit RAD21 by Hepatitis C virus mediates host-virus interactions. Nucleic acids research 47: 2455-2471
Reviewer 2 Report
HCV infection is the risk factor for hepatocellular carcinoma (HCC) and leads to a more aggressive and invasive diseases. The virus induces metastatic spreading of HCC tumors through regulation of invadopodia. Here authors demonstrated that NS3/4A protease inactivates T cell protein tyrosine phosphatase (TC-PTP) resulting in sustained activation of EGFR signaling. This leads to upregulation of invadopodia-associated genes including matrix metalloproteases.
Gene expression analysis, and in vitro and in vivo experiments support their conclusion well. HCV-infection affected intrahepatic and extrahepatic dissemination of tumor cells but not their proliferation by inducing invadopodia formation.
The authors infected Huh7.5 tumor cell line with HCV and showed that the infection induced dissemination of Huh7.5. The reviewer is wondering whether HCV infection increases migration or invasiveness of the healthy hepatocytes. HCC formation in a HCV-patient is the long process; hepatitis, fibrosis, and cirrhosis. The reviewer assumes that dissemination is the very last step of disease progression. If HCV specifically induces invasiveness of tumor cells, it is interesting to consider the reason why HCV affects healthy hepatocytes and tumor cells in a different manner. It would be better to argue this point in discussion.
Author Response
Point 1: The authors infected Huh7.5 tumor cell line with HCV and showed that the infection induced dissemination of Huh7.5. The reviewer is wondering whether HCV infection increases migration or invasiveness of the healthy hepatocytes. HCC formation in an HCV-patient is the long process; hepatitis, fibrosis, and cirrhosis. The reviewer assumes that dissemination is the very last step of disease progression. If HCV specifically induces invasiveness of tumor cells, it is interesting to consider the reason why HCV affects healthy hepatocytes and tumor cells in a different manner. It would be better to argue this point in discussion.
Response 1: We thank the reviewer for this comment. Indeed, we agree with the reviewer that an important and interesting point is whether HCV infection increases invasiveness of healthy hepatocytes or only of tumor cells, and should be discussed. In principle, to initiate the metastatic process, cells must first acquire motility and invasive abilities, allowing their transport to distant sites or organs. At secondary sites, carcinoma cells can form a new carcinoma through a mesenchymal—epithelial transition (MET), in which cells acquire the ability to proliferate (Murphy & Courtneidge, 2011). As previously shown by Akkari et al., (Akkari et al, 2012), expression of HCV proteins in primary hepatic precursors and in immortalized hepatocyte cell lines induced EMT and cell migration and invasion in vitro. However, in mice xenograft model, where the cells expressing viral proteins were injected sub-cutaneously, no tumors were observed. Importantly, the addition of Ras following injection of HCV proteins-expressing cells induced extensive tumor formation and metastasis in the lungs. These results indicate that indeed infected primary hepatocytes do acquire migratory and invasive properties which support our hypothesis that the properties of invasion of the infected cells may provide an advantage for HCV for its survival and spread, also at initial stages of infection, as suggested in the discussion. However, in primary hepatocytes, the presence of HCV may not be sufficient for tumor formation and metastasis, and additional factors that are produced by tumor cells, such as Ras, are required for the proliferation of the invasive cells which leads to enhanced aggressiveness and metastasis formation in vivo. Collectively, this suggests that metastatic process in HCV infected cells occurs in late stage following the progression to cancer. Therefore, we have demonstrated the HCV-induction of invasion and metastasis in HCC cells.
This point was included in the discussion line 725 of the revised manuscript.
References
Akkari L, Gregoire D, Floc'h N, Moreau M, Hernandez C, Simonin Y, Rosenberg AR, Lassus P, Hibner U (2012) Hepatitis C viral protein NS5A induces EMT and participates in oncogenic transformation of primary hepatocyte precursors. Journal of hepatology 57: 1021-1028
Murphy DA, Courtneidge SA (2011) The 'ins' and 'outs' of podosomes and invadopodia: characteristics, formation and function. Nature reviews Molecular cell biology 12: 413-426
Reviewer 3 Report
The paper explains that HCV infection doesnot impact cell migration potential however it promotes invasion capacity using EGFR pathway.
Major Comments
The study is based on RNAseq data mostly. In RNAseq analysis its shown induction of DDR1/2 genes and protooncogene RET signifiantly more that EGFR. Impact of DDR1/2 and RET deletion should be done in comparison to EGFR, performing cell invasion assay. How important is MMP14 compared to other MMPs. Show MMPs expression profiles distinctly (qPCR). MMP1, MMP2, MMP9 and compare with MMP14. HCV can promote collagen deposition in liver and out side which can easily activate DDR1/2 and that can prmote MMP activation and cell invasion. Link the upstream signal with MMP activation. Most of the genes related to invasion also plays important role in cell migration. If cell migration is completely unaffected ( show more proves of different experiments of Boyden chember or live cell imaging) then it mostly due to proteases that helps degrading ECM. So, emphasis should be given on them in title rather that inavadopodia. Most of the images are very low quality. Westernblots are poor quality. Author must consider presenting good quality of figures. Author should address the above points to be reconsidered.
Author Response
Point 1: The study is based on RNAseq data mostly. In RNAseq analysis its shown induction of DDR1/2 genes and protooncogene RET significantly more than EGFR. Impact of DDR1/2 and RET deletion should be done in comparison to EGFR, performing cell invasion assay.
Response 1: We appreciate the reviewer’s thoughtful note on this issue. DDR1/2 and RET are not found within the differentially expressed genes in our RNAseq data, however they are significantly activated as demonstrated by the Pamgene kinase activation assay in Figure 4. Indeed, both DDR proteins and RET are activated in the assay and RET shows more significant signal in the graph in Figure 4B.
Although RET and DDR proteins are known to be involved in invasiveness, to our knowledge they have not shown direct connection to invadopodia, whereas EGFR was shown by us and others to directly induce formation and activation of invadopodia (Mader et al, 2011). Therefore, because the main point of the manuscript is the effect of HCV on invadopodia, and because HCV uses EGFR as a co-receptor for entry into the host cell while activating it, and also because EGFR is the protein with highest connectivity (Figure 4C) we focused on EGFR in this specific manuscript. We feel that concentrating on DDR and RET will shift the focus and main point of this manuscript, and will be beyond the scope of this specific story.
Point 2: How important is MMP14 compared to other MMPs. Show MMPs expression profiles distinctly (qPCR). MMP1, MMP2, MMP9 and compare with MMP14.
Response 2: We thank the reviewer for this constructive comment. MMP14 (MT1-MMP) is a membranal MMP that activates secreted MMPs such as MMP2 and MMP9, and is directly related to invadopodia maturation. MMP14 appeared in the overlaps of invasion-RNAseq (Figure 1E) and invadopodia-RNA-seq (Figure 2A) and its increased expression in HCV infected cells is also validated by qRT-PCR (Figure 2C). Therefore, we anticipated that this MMP will have a significant role in HCV-induced increased matrix degradation, as was validated in Figure 3C. The MPPs MMP1, MMP2, and MMP9 were not detected to be altered by our screens and therefore were not tested in infected vs. non-infected cells. However, since MMP2 and MMP9 are also activated upon invadopodia maturation, we agree with the reviewer that it is important to show expression of these MMPs compared to MMP14 in infected compared to non-infected cells. Therefore, as requested by the reviewer, expression differences of MMP2, MMP9 were tested by qPCR, and presented in Supplementary Figure 3 in the revised manuscript. As shown in Supplementary Figure 3, a significant increase in expression of all three MMPs was observed, however, MMP14 showed the highest increase in expression following infection, validating its significant role in HCV-induced cell invasiveness compared to other MMPs.
We added explanations in line 426 in the revised manuscript.
Point 3: HCV can promote collagen deposition in liver and outside which can easily activate DDR1/2 and that can promote MMP activation and cell invasion. Link the upstream signal with MMP activation.
Response 3: We appreciate the thoughtful comment of the reviewer on this topic. MMPs are mainly secreted from invadopodia, which are matured and activated by the action of receptor and non-receptor tyrosine kinases such as EGFR, Arg, Src, Pyk2. Interestingly, all kinases showed increased activation in the Pamgene kinase activation assay (Figure 4). While the link between MMP activation and/or ECM degradation by invadopodia has been previously demonstrated in publications from the Gil-Henn laboratory (Genna et al, 2018; Mader et al, 2011), a direct link between DDR1/2 and invadopodia in cancer cells has never been demonstrated. The authors feel that a discussion linking DDR and MMPs will be inappropriate and beyond the focus of this specific manuscript.
Point 4: Most of the genes related to invasion also play important role in cell migration. If cell migration is completely unaffected (show more proofs of different experiments of Boyden chamber or live cell imaging) then it mostly due to proteases that help degrading ECM. So, emphasis should be given on them in title rather than invadopodia.
Response 4: We appreciate the reviewer’s thoughtful comment on this issue. It is now well known that invasion and migration are two different and separate processes (Genna & Gil-Henn, 2018; Genna et al, 2018). Figure 1 shows no difference in cell migration, while a significant difference in invasion, which is correlated with increase in invasion and invadopodia genes is observed. In addition, as explained above, MMPs in cancer cells are mainly secreted from invadopodia, which goes along with the suggested model in Figure 8.
As requested by the reviewer, we have included an additional proof that cell migration is unaffected by HCV infection, by performing a Boyden chamber experiment that is presented in Supplementary Figure 2.
Point 5: Most of the images are very low quality. Western blots are poor quality. Authors must consider presenting good quality of figures.
Response 5: As requested by the reviewer, we have replaced the figures in the manuscript document with higher quality figures in the revised manuscript. Moreover, original images and figures with higher quality are uploaded to the journal website.
Additionally, and as required by the reviewer, we have repeated the experiment presented in Figure 5F-G and replaced the blot by a higher quality blot in the revised manuscript.
References
Genna A, Gil-Henn H (2018) FAK family kinases: The Yin and Yang of cancer cell invasiveness. Molecular & cellular oncology 5: e1449584
Genna A, Lapetina S, Lukic N, Twafra S, Meirson T, Sharma VP, Condeelis JS, Gil-Henn H (2018) Pyk2 and FAK differentially regulate invadopodia formation and function in breast cancer cells. The Journal of cell biology 217: 375-395
Mader CC, Oser M, Magalhaes MA, Bravo-Cordero JJ, Condeelis J, Koleske AJ, Gil-Henn H (2011) An EGFR-Src-Arg-cortactin pathway mediates functional maturation of invadopodia and breast cancer cell invasion. Cancer Res 71: 1730-1741
Round 2
Reviewer 3 Report
Can be accepted with a minor revision. In Supl Fig 2, in migration data, looks like HCV infected cells migrated more than non infected. Please give a statistics and a violin plot to see the impact of cell migration in most of the cells. Also, please show in fold difference wise the effect of infection on cell invasion vs migration.
Author Response
Point 1: Can be accepted with a minor revision. In Supl Fig 2, in migration data, looks like HCV infected cells migrated more than non infected. Please give a statistics and a violin plot to see the impact of cell migration in most of the cells. Also, please show in fold difference wise the effect of infection on cell invasion vs migration.
Response 1: We agree with the reviewer that the HCV infected cells migrated more the non-infected cells, but this difference was not statistically significant (P=0.0641). We have changed the main text accordingly, in line 303: ”To validate these results we also performed a Transwell migration assay. We plated serum-starved HCV-infected or non-infected Huh7.5 HCC cells on Transwell membranes and measured their ability to migrate towards complete medium. We observed higher number of migrated cells in HCV infected compared to non-infected cells, however, the difference was not statistically significant (Figure S2A)“
As requested by the reviewer, we changed the graphics in Supplementary Figure 2A to a violin plot in the revised manuscript. We also provided the statistics in the supplementary figure 2A legend (P=0.0641).
Also, according to the reviewer’s request, we added supplementary Figure 2B that illustrate the effect of infection on cell invasion vs migration in fold difference. This comparison pronounces the significant effect of HCV infection on invasion vs. the non-significant effect on migration. We thank the reviewer for this constructive comment.